# Biological Applications of Thiocarbohydrazones and Their Metal Complexes: A Perspective Review

**DOI:** 10.3390/ph13010004

**Published:** 2019-12-25

**Authors:** Carmela Bonaccorso, Tiziano Marzo, Diego La Mendola

**Affiliations:** 1Dipartimento di Scienze Chimiche, Università degli Studi di Catania, Viale A. Doria 6, 95125 Catania, Italy; 2Dipartimento di Farmacia, Università di Pisa, Via Bonanno Pisano 6, 56126 Pisa, Italy; tiziano.marzo@unipi.it

**Keywords:** thiocarbohydrazones, metallodrugs, anticancer, antimicrobial, metal ions

## Abstract

Although organic compounds account for more than 99% of currently approved clinical drugs, the established clinical use of cisplatin in cancer or auranofin in rheumatoid arthritis have paved the way to several research initiatives to identify metal-based drugs for a wide range of human diseases. Nitrogen and sulfur donor ligands, characterized by different binding motifs, have been the subject in recent years of one of the main research areas in coordination chemistry. Among the nitrogen/sulfur compounds, very little is known about thiocarbohydrazones (TCH), the higher homologues of the well-known thiosemicarbazones (TSC), and their metal complexes. The extra hydrazine moiety provides the ligands of variable metal binding modes, structural diversity and promising biological implications. The interesting coordination chemistry of TCH has mainly been focused on symmetric derivatives, which are relatively simple to synthesize while few examples of asymmetric ligands have been reported. This informative review on TCHs and their metal complexes will be helpful for improving the design of metal-based pharmaceuticals for applications ranging from anticancer to antinfective therapy.

## 1. Introduction

The use of metals for medicinal purposes has been exploited since very ancient times; for instance, silver has been used as a disinfectant agent for water and milk for thousands of years [1]. Similarly, the application of gold in medicine can be dated back to 2500 BC and throughout the entire history of humanity it is possible to find traces of several applications of this noble metal to treat various diseases [2]. Indeed, in the 19th century the complex dicyanoaurate(I) (K[Au(CN)_2_]) was proposed by Koch for its bacteriostatic properties to fight tubercle bacillus, while in the 20th century, gold complexes were introduced to treat rheumatoid arthritis, leading to the approval of Auranofin by FDA in 1985. This latter compound is today the reference compound for gold complexes and, on the ground of the so-called repurposing strategy, has been proposed as promising anticancer agent and entered several clinical trials in US, some of which are still ongoing [3,4].

Beyond silver and gold, several bismuth, antimony and mercury compounds have been employed to combat bacterial and parasitic diseases. In this view, bismuth salts and antimony complexes have been proposed and used for eradication of Helicobacter pylori infection, and against leishmaniasis respectively [5]. Also, arsenic, in the form of trioxide (As_2_O_3_), is nowadays one of the reference drugs for the treatment of acute promyelocytic leukemia [6]. Several inorganic complexes are also used for diagnostic medicine as in the case of gadolinium-based contrast agents, ^99m^Tc compounds for myocardial perfusion imaging [7] and ^64^Cu for PET imaging [8].

However, the most important impulse to the research of metal-based drugs with medicinal properties came from the serendipitous discovery of the antitumor features of cisplatin by Rosenberg and Loretta Van Camp in 1965 [9]. Cisplatin was approved in 1978 by the FDA, and this event triggered enormous efforts by scientists in search of innovative and ameliorated inorganic anticancer drugs, leading to the approval worldwide of carboplatin and oxaliplatin analogs. This makes platinum-based drugs an essential arsenal for first- and second-line anticancer chemotherapy used in about 50% of clinical protocols [10].

Metals complexes offer an extremely versatile and reliable tool for the development of improved medicinal compounds. Indeed, it is possible to finely tune the chemical properties of these complexes by controlling the metal center oxidation state and selecting the most appropriate ligands for each application. Thus, it is not surprising that the challenge in the development of innovative and improved metal-based drugs largely overlaps with the development of innovative and ameliorated ligands for the functional metal element.

Among several ligands, thiosemicarbazones (TSCs), are a very attractive class of metal-chelating ligands, able to coordinate many transition metals through the sulfur as well as the azomethinic nitrogen atoms [11]. They can act as N, S-multidentate ligands, and moreover, it is possible to modulate the binding properties/stoichiometries through the insertion of other heteroatoms into the backbone structure (i.e., phenolic or pyridyl moieties). They have a great variety of biological properties, both as free ligands and as metal complexes, and several studies have been published reporting on thiosemicarbazone-based complexes with medicinal applications [12,13,14,15,16,17,18,19,20,21].

Beyond TSCs, in recent years, there has been growing interest in the coordination chemistry of thiocarbohydrazones (TCHs) compounds that share the general formula depicted in Scheme 1a–c and that can be considered the higher homologues of TSCs (Scheme 1d). The first synthesis of these systems is dated 1925 and described the condensation of ketones and aldehydes with thiocarbohydrazide [22]. The earliest employ of this derivatives relied the hetero-ring closure of the aldehyde derived thiocarbohydrazones [22,23,24].

While TSC started to be used in the late sixties [25], the first report on the “*Preparation and fungistatic properties*” of neat TCH appeared only in the following decades [26], probably because the key thiocarbohydrazide intermediate was not commercially available until the late 1970s and should be synthesized from carbon sulfide and hydrazine. Until the first half of the nineties, however, most reports concerned the synthesis and characterization of ligands and their complexes with transition metals were reported. Only recently it was understood the potential of these systems: TCHs are able to act as metal-based drugs and their biological activities were investigated.

The analytical uses of TCHs as spectrophotometric reagents in metal determination has been reviewed in recent years [27] and is continuously developing. However, despite their ability to coordinate metals and the remarkable biological properties of the resulting complexes, no general survey on the employment of TCHs as suitable ligand for the synthesis of metal-based drugs has yet reported. Here we give an overview of TCHs structures and tautomerism; the different biological activities as antibacterial, antimicrobial and anticancer agents of their metal complexes have been reported. Due to the huge variety of metal cations employed in biologically active TCH complexes (eighteen cationic species, ranging from the first-row transition metal cation to lanthanides, from organo-metal to oxo-metal cation, will be reported in this review) and to the different binding stoichiometries available for each cation, we decided to make use of a simpler classification based on the ligand backbone: the biological activity of the metal complexes derived from symmetric, macrocyclic and asymmetric TCHs has been discussed in three different section. Further sub-sections have been employed to describe the most common substituents used in the TCHs ligand synthesis (R_1_, R_2_, R_3_ and R_4_ groups in Scheme 1), namely salycilic, pyridyl, aromatic/heteroaromatic, ferrocenyl, isatin, coumarin, (and related) carbonyl compounds.

For the same reasons, together with the novelty represented by these systems and the numerous fields of application, it is not possible to identify a unique action mechanism: each combination (metal cation/TCHs backbone/field of employ) can give rise to a different mechanism of action, which has only been deeply investigated in a few cases; when available, clear indications of the action mechanism of the TCHs metal complexes are reported in the text.

## 2. Structures of Thiocarbohydrazones

From a structural point of view, we can distinguish between symmetric and asymmetric systems. In the first case, the condensation of the thiocarbohydrazide with two equivalents of aldehyde or ketone yields 1,5-bisthiocarbohydrazones (Scheme 1a); in the latter case, the condensation with one equivalent of the carbonyl derivative is exploited, leaving a terminal non-functionalized hydrazine moiety (Scheme 1b) for the subsequent insertion of a different carbonyl compound (Scheme 1c).

As in the case of the parent thiocarbohydrazides, the thioketo-thioenol tautomerism is also possible for the thiocarbohydrazones, see Scheme 2. Moreover, the thioenol tautomers can exist as *syn* (Scheme 2b) or *anti* (Scheme 2c) geometric isomers as a consequence of the double bond character of the central N-C linkage.

TCHs constitute interesting ligand systems because of the availability of several potential donor sites; they usually act as neutral or negatively charged ligands and bind the metal through the sulfur atom and one imine nitrogen atom. Such behavior is a consequence of the tautomeric equilibrium between the thioketo and the thioenol forms. Several factors affect this equilibrium, namely the nature of the metal ion and its counterion, the reaction conditions, the nature of the solvent and the pH of the medium. The presence of additional coordination sites on the lateral substituents can influence both the stoichiometry and the selectivity of the binding with the different metal ions.

These features can be exploited in different fields of biological application, and the possibility to finely tune the chelate stability/properties by modifying the coordinating residues of the substituents may have very important results. So far, only a few reports on the biological activity of TCHs are available, being tested for medicinal applications only sporadically with respect to their potential antiviral activity against herpes through induction of HSV-1 ribonucleotide reductase inactivation [28]. Similarly, while thiosemicarbazones have been extensively exploited for the synthesis of metal complexes with anticancer, antimicrobial or antifungal applications [29,30], thiocarbohydrazone complexes have received far less attention. This is something of quite surprisingly due to the very strict similarities between these two types of ligands, able to coordinate metal centers in a multidentate fashion. For instance, only few examples of TCHs-based metal compounds with Fe(III), Mn(II), Cr(III), Co(II), Ni(II), Cu(II) or Zn(II) centers were tested for their pharmacological properties against cancer, bacterial strains or fungal infection [31,32]. In this context, no systematic biological investigations have been made on this class of metal compounds that remain overlooked in their possible medicinal applications.

## 3. Symmetric 1,5-Bisthiocarbohydrazones

The symmetric thiocarbohydrazones can be readily synthesized through a one-pot method by refluxing thiocarbahydrazide with aldehydes/ketones (in slight excess) in alcohol or alcoholic mixtures; in some cases, is reported the presence of acetic acid as a catalyst. The solid product separates on cooling; the TCHs thus obtained are usually highly crystalline and were initially suggested to be useful for characterizing aldehydes and ketones [33]. The 1,5-bisthiocarbohydrazones derived from salicylaldehyde (and related carbonyl compounds) are no doubt the most widely studied members of this compound family, for both their complexation ability and biological properties. Of great interest in the last decade have been systems derived from pyridyl and related compounds, while other aromatic/heteroaromatic derivatives have been less successful. Several TCHs derived from natural compounds, such as coumarin, isatin, cholesterol etc., have also been reported.

### 3.1. Symmetric Thiocarbohydrazones Derived from Salycilaldehyde and Related Carbonyl Compounds

The first report on the synthesis of symmetrical TCHs obtained using the salicylaldehyde or its derivatives dates back to 1925 [22]. However, for the first data on the biological activity of these ligands, and the corresponding metal complexes, it is necessary to wait until 1985, when Patil et al. reported the synthesis, characterization and bactericidal activity of both mono (Co(II) and Cu(II)) and bimetallic (Ni(II) plus Cu(II)) complexes of the thiocarbohydrazones (1, Scheme 3) [34]. Antibacterial and antifungal activities are the main properties studied for these classes of TCHs, but the presence of the phenolic ring has made it possible to exploit them also for their antioxidant capacities.

Good antibacterial activity against *E. coli* bacteria, together with fluorescence and pigmentation studies, was reported ten years later for the TCHs derived from salicylaldehyde, o-vanillin, 2,4-dihydroxybenzaldehyde or 2-hydroxy-1-naphthaldehyde (2, Scheme 3) and their twelve metal complexes of Cu(II), Ni(II) and Zn(II), isolated as [M_2_L_2_(H_2_O)_4_] [35]. Interestingly, the ligands with methoxy groups show enhanced activity, but the metal complexes are more toxic in comparison to the parent ligands. The toxicity of the complexes is inversely correlated with the radius of the metal ion (i.e., Ni, Cu, Zn) as the decreases in polarizability of the metal center could enhance the lipophilicity and permeability of the complexes and, thus, the interference with the normal cell processes.

After these pioneering studies, in 2004 the synthesis, characterization and antimicrobial studies on bacteria and fungi were reported for both new bis(thiocarbohydrazone) ligands (3, Scheme 3), derived from 2-hydroxyacetophenone and 5-chlorosalicylaldehyde, and two series of their transition metal complexes with Cr(III), Mn(II), Fe(III), Co(II), Ni(II), Cu(II) and Zn(II) ions [32]. The thiocarbohydrazone ligands and their transition metal complexes were screened for antibacterial and antifungal activity using the bacteria *Azotobacter* and *Rhizobium* and the fungus *Fusarium oxysporium*. The values of the percentage of inhibition of bacteria or fungal growth show that all the complexes are more active than their thiocarbohydrazone ligands.

Over the past 10 years, several groups have employed these derivatives and related metal complexes. Shebl et al. reported a new ligand formed by the condensation of thiocarbohydrazide with 2-hydroxy-1-naphthaldehyde (4, Scheme 3) and a series of binary complexes of Fe(III), Co(II), Ni(II), Cu(II), Zn(II), Ce(III) and U(VI) [36]. Ternary complexes have also been synthesized by using 1,10-phenanthroline or oxalic acid as a secondary ligand. The ligand and its complexes were investigated as antimicrobial agents against the sensitive organisms *S. aureus* as Gram-positive bacteria, *E. coli* as Gram-negative bacteria and the fungi *C. albicans* and *A. flavus*. Both the newly prepared TCH and some binary metal complexes showed a remarkable activity against *E. coli* and *S. aureus*. Moreover, a ternary complex with Co(II) and 1,10-phenanthroline showed a higher activity than amphotricine B that was used as a control antifungal agent. As previously reported, the lipophilicity controls the antimicrobial activity as it enhances the penetration of ligand and its metal complexes into the lipid membranes and thus restricts further growth of the organism.

Few works have been reported on organotin(IV) complexes with thiocarbohydrazone, for this reason, the work of Chee et al. is noteworthy, focusing on six new complexes obtained by direct reaction of RSnCl_3_ (R = Me, Bu and Ph) or R_2_SnCl_2_ (R = Me, Bu and Ph) and 2-hydroxyacetophenone thiocarbohydrazone (3a, Scheme 3) [37]. They were screened for antibacterial activity against *Escherichia coli*, *Staphylococcus aureus*, *Salmonella typhi* and *Enterococci aeruginosa*. Both the ligand and the organotin(IV) complexes show weak to moderate activity against tested bacteria; only the diphenyl and dibutyl derivatives showed good antibacterial activity against *S. aureus* and *E. aeruginosa*. The parent TCH also gives weak inhibition towards *St. aureus* growth. More recently, three new organotin(IV) complexes were formed from the reaction of R_2_SnCl_2_ (R = Ph, Me and Bu) with 1,5-bis((2-hydroxynaphthaldehyde) thiocarbohydrazone (2d, Scheme 3) [38]. The in vitro antibacterial activities of the ligand and the organotin(IV) complexes, against *Bacillus subtilis* and *Staphylococcus aureus* (as Gram-positive bacteria) and *Escherichia coli* and *Pseudomonas aeruginosa* (as Gram-negative bacteria), were compared to standard antibacterial drugs, viz., vancomycin, streptomycin, penicillin, nalidixic acid, and gentamycin. The DNA cleavage activity was also investigated. The results showed that the ligand did not have antibacterial effect against all tested bacteria. It also did not show DNA cleavage activity against either chromosomal or plasmid DNA. The organotin(IV) complexes inhibited bacterial growth, showing both plasmid and chromosomal DNA cleavage activity; this cleavage activity was therefore related to the ability of the complexes to diffuse into bacterial cytoplasm, thanks to the increased membrane fluidity: the chelation of the metallic cations enhances the hydrophobicity of the systems and the subsequent insertion of complexes into bilayer membrane affect lipid packaging. The non-covalent interactions of the metal complexes with DNA (intercalation and groove binding) cleave the duplex, cause DNA damage, block the division of cells, and result in cell death.

The di-substituted halogen 1,5-bisthiocarbohydrazones were obtained by the condensation of 3,5-dibromosalicylaldehyde, 3-bromo-5-chlorosalicylaldehyde and 3,5-dichlorosalicylaldehyde with thiocarbohydrazide (5, Scheme 3) [39]; their dioxomolybdenum(VI) complexes formed a labile coordination site that can be used for substrate binding. The ternary complexes obtained with reagents such as DMSO, DMF, pyridine, methanol, ethanol have been screened for their antioxidant capacity, by using the cupric reducing antioxidant capacity (CUPRAC) method, and the antimicrobial activity against *Staphylococcus aureus*, *Staphylococcus epidermidis*, *Escherichia coli*, *Klebsiella pneumoniae*, *Pseudomonas aeruginosa*, *Proteus mirabilis* and *Candida albicans*. The ligands and complexes showed good antioxidant capacity, moreover the complexation significantly enhances the properties of the tested compounds. When compared to antimicrobial drugs ciproflaksasin and flukanazol, all compounds were found to be completely inactive against the reported bacterial strains. Neutral solvate dioxomolybdenum(VI) complexes with the general formula [MoO_2_L(ROH)] were prepared by the reaction of 2-hydroxybenzophenone, 2-hydroxy-4-methoxybenzophenone, 2-hydroxy-4-octyloxybenzophenone, 2-hydroxy-4-methoxy-4′-methylbenzophenone and 2-hydroxy-4-allyloxybenzophenone with thiocarbohydrazide (6, Scheme 3), followed by addition of MoO_2_(acac)_2_ with the proper alcohol (methanol, ethanol and n-butanol) being used as solvent [40]. The ligands and complexes, tested for in vitro antioxidant capacities, show higher antioxidant activities when compared to the standard (Trolox) and also, in this case, the activities of the complexes are higher than the ligands.

### 3.2. Symmetric Thiocarbohydrazones Derived from Pyridyl and Related Carbonyl Compounds

The study of the biological properties of symmetric pyridyl-based TCHs has only recently been developed, despite the numerous structural studies both on ligands and on complexes with transition metals [41,42,43,44,45,46,47,48]. Numerous studies have shown that these systems exhibit different chemical and biological properties. Bacchi et al. reported the synthesis of a series of new mono- and bisthiocarbonohydrazone ligands (7, Scheme 4) and their Cu(II), Fe(II) and Zn(II) complexes: the in vitro antimicrobial activity was exhaustively evaluated against bacteria and fungi [49]. The microorganisms tested as Gram-positive bacteria were: *Bacillus brevis*, *Bacillus cereus*, *Bacillus circulans*, *Bacillus megaterium*, *Bacillus pumilus*, *Bacillus subtilis*, *Bacillus subtilis*, *Bacillus thuringiensis var. finitimus*, *Bacillus thuringiensis var. kurstaki*, *Sarcina lutea*, *Staphylococcus aureus*, and clinical isolates of *Staphylococcus epidermidis* and *Streptococcus faecalis*. The Gram-negative bacteria used were *Enterobacter cloacae*, *Escherichia coli*, *Proteus vulgaris*, *Pseudomonas aeruginosa*, *Pseudomonas cepacia*, *Salmonella typhimurium*, *Serratia marcescens* and clinical isolates of *Klebsiella pneumoniae* and *Proteus mirabilis*. The yeasts employed were *Candida tropicalis* and *Saccharomyces cerevisiae*, while *Aspergillus niger* was used as a mould. Some of the bisthiocarbonohydrazones described in this paper are remarkable antimicrobial agents against Gram-positive bacteria, as they are endowed with high antibacterial activity and lack mutagenic properties. As previously reported for the symmetric TCHs derived from salycilaldehyde, the results of this study indicate the importance of lipophilicity for the antimicrobial activity of thiocarbonohydrazones ligand. Surprisingly, the metal complexes exhibit inhibitory activity comparable to or lower than that of the starting ligands. Relevant exceptions are some Cu(II) complexes, because, among the investigated complexes, these derivatives show higher antimicrobial potency than the corresponding Fe(II) and Zn(II) derivatives.

The bis(pyridine-2-methylidene)-thiocarbohydrazone (7c, Scheme 4) and its Ni(II), Co(II), Cu(II), and Zn(II) complexes were also used to study the DNA binding mechanism and binding constants in an aqueous medium by absorption spectroscopy by using Escherichia coli DNA as target [50]. The trans-geometrical isomers of the complexes give reason of an improved interaction of the metal complexes (the binding constants follow the order of Ni(II) ≥ Co(II) > Zn(II) > Cu(II)) compared to ligand isomers in cis-positions. The Ni(II) and Co(II) complex showed the best interaction with DNA due to both the possible geometry of metal coordination to the TCH ligand (distorted tetrahedral in the former case and square pyramidal in the latter; they were indicated as trans-geometrical isomers) and the availability of the vacant d-orbital. The strong stacking interaction between the aromatic chromophore and the base pairs define the DNA binding property of the metal complexes: the Ni(II) and Co(II) complexes acts as groove binder with the DNA. The Cu(II) and Zn(II) complexes have much less binding affinity; they even have different binding modes of interaction or bonding to the DNA.

The tridentate thiocarbahydrazones derived from 2-acetylpyrazine (8, Scheme 4) and their Bi(III), Ga(III) and diorganotin(IV) complexes exhibit interesting properties structurally and biologically [51]. Both the free ligand and the complexes are capable of inhibiting cell proliferation growth and could slightly distinguish the human hepatocellular carcinoma HepG2 cells from normal hepatocyte QSG7701 cells. The coordination to Bi(III) might be an interesting strategy for increasing cytotoxicity as this complex promoted a dose-dependent apoptosis in HepG2 cells that was associated with an increase in intracellular ROS production and reduction of mitochondria membrane potential (MMP).

Quinoline is a recurring scaffold in anticancer drug discovery and various quinoline thiocarbohydrazones (9–10, Scheme 4) were tested against cancer cell lines showing promising results on two diverse cell lines, acute monocytic leukemia (THP-1) and pancreatic adenocarcinoma cancer stem cells (AsPC-1), both in a 2D monolayer model and in spheroidal 3D culture [52]. These compounds target different phases of mitotic division in a concentration-dependent manner, and two bis-TCHs revealed pro-apoptotic activity on CSCs. The activation of caspase-8 indicated an endoplasmic reticulum stress or even activation of death receptor signaling, while no clear correlation emerged between potency in ROS production and intensity of apoptotic response. These results pave the way to further investigations of mono- and bis-TCHs with mechanism of caspase-8 activation as the primary interest. Cu(II) complexes of 1,5-bis(quinoline-2-carbaldehyde) thiocarbohydrazone (9a, Scheme 4) were analyzed for anticancer activity in breast cancer cell lines [53] and exhibited antiproliferative effect as assessed by MTT assay in MCF-7 and MDA-MB-231 cells. Copper complexation is essential for cytotoxicity as the ligands alone induced cell proliferation.

Recently, Božic et al. explored the antimicrobial activity of twenty-two mono- and bis- thiocarbohydrazones derived from 2-acetylpyridine and related carbonyl compounds (9–11, Scheme 4) [54]. The antibacterial activity was evaluated using four different strains of Gram-positive bacteria (*Bacillus subtilis*, *Staphylococcus aureus*, *Clostridium sporogenes* and *Kocuria rhizophila*) and four different strains of Gram-negative bacteria (*Escherichia coli*, *Pseudomonas aeruginosa*, *Proteus hauseri* and *Salmonella enterica subsp. Enterica serovar Enteritidis*). Interestingly, a 3D QSAR analysis, based on molecular interaction fields, highlighted the structural fragments important for biological activity towards the different strains. The authors also suggested several structural modifications that could improve the potency of the TCHs, the absorption and distribution in the body, along with reduced toxicity.

### 3.3. Symmetric Thiocarbohydrazones Derived from Aromatics/Heteroaromatics and Related Carbonyl Compounds

Although the first synthesis of TCHs, reported in 1925 [22], concerned the condensation of thiocarboidrazide with aromatic/heteroaromatic aldehydes and ketones (12, Scheme 5), it is necessary to wait until the 1970s for the first report on the fungistatic properties of some benzyl-derived ligands, and it is only at the dawn of the new millennium that the use of metal complexes gains attention [26]. However, the absence of additional coordination sites, in addition to the central TCH unit, restricts the type and the number of metal complexes accessible, unlike that reported for salicylic and pyridyl derivatives. Thiocarbohydrazones derived from different benzaldehydes, acetophenones and benzophenone have also attracted recent interest in the field of anion sensing [55,56,57].

Different TCHs derived from pentatomic heterocycles have been reported by Bacchi et al., together with some pyridyl derived systems (13, Scheme 5): the new ligands and their Cu(II), Fe(II) and Zn(II) complexes were evaluated in vitro for the antimicrobial activity against Gram-positive and Gram-negative bacteria, yeasts and fungi [49]. The antibacterial and antifungal activities of the newly synthesized thienyl derivatives and their La(III) and Th(IV) complexes (14, Scheme 5) were studied by Prakash et al. [58]. The ligand is weakly active against *E. coli*, *S. aureus* (bacteria) and highly active against *A. niger* and *A. fumigates* (Fungi); whereas its La(III) complex shows high activity against *S. aureus* (bacteria), *A. niger* and *A. fumigates* (Fungi). On the other hand, the Th(IV) complex shows more activity against *P. auregonosa* (bacteria), *A. niger* and *A. fumigates* (fungi).

### 3.4. Symmetric Thiocarbohydrazones Derived from Isatin, Coumarin and Related Compounds

Isatin and its derivatives represent an important class of heterocyclic compounds frequently employed in drug synthesis, included TCHs and their metal complexes. In a direct comparison with the TSCs homologues, in 2005 it was reported the synthesis and characterization of a new ligand 1,5-bis(isatin)thiocarbonohydrazone, of its N-alkyl derivatives (15, Scheme 6) and their diorganotin(IV) monometallic complexes [59]. All the compounds were tested for antimicrobial and mutagenic properties. Antimicrobial testing was performed against Gram-positive bacteria (*Bacillus cereus*, *B. circulans*, *B. megaterium*, *B. pumilus*, *B. subtilis*, *B. subtilis var. natto*, *B. thuringiensis var. kurstaki*, *Sarcina lutea*, *Staphylococcus aureu*, *S. epidermidis* and *clinical isolate of Streptococcus fecalis*), Gram-negative bacteria (*Enterobacter cloacae*, *Escherichia coli*, *Klebsiella pneumoniae*, *Proteus vulgaris*, *Pseudomonas aeruginosa*, *Salmonella typhimurium*, *Serratia marcescens*), yeasts (*Candida tropicalis*, *Saccharomyces cerevisiae*) and moulds (*Aspergillus niger*). The isatin and N-methylisatin organotin(IV) complexes exhibit the highest antibacterial activity against Gram-positive bacteria and some of the butyl-tin compounds are broad spectrum agents as they are active also towards Gram-negative bacteria. Moreover, the antibacterial properties of these complexes are coupled with lack of mutagenicity. The synthesis and characterization of the Co(II), Ni(II), Cu(II), and Zn(II) complexes of TCH ligand 15a were reported by Sathisha et al. [31]. The in vitro short-term cytotoxic activity was determined using Ehrlich Ascites Carcinoma cells and in vivo using the tumor model in mice: all the compounds were found to have considerable cytotoxicity in the cell viability test. The antibacterial activity was assessed against *Bacillus cirroflgellosus*, while *Aspergillus niger* and *Candida albicans* were used to test the fungicidal activity. The Co(II) complex showed promising results against both the fungi and, overall, the metal complexes exhibited higher fungi toxicity.

The coumarin ring system also displays interesting pharmacological properties that trigger the synthesis of the symmetric TCH derived from 4-hydroxycoumarin (16, Scheme 6) and the binuclear complexes with Cu(II), Ni(II), Zn(II), Co(II), Mn(II), Fe(III) and Cr(III) ions, with the object of gaining more information about their nature of coordination and related structural and spectral properties as well as their antimicrobial properties [60]. The ligand and its metal complexes were evaluated for antimicrobial activity against one strain Gram-positive bacteria (*Staphylococcus aureus*), Gram-negative bacteria (*Escherichia coli*), and fungi (*Candida albicans* and *Fusarium solani*). As previously seen, the resulting metal complexes enhanced the antimicrobial activity of the free ligand: the increased activity observed upon chelation is related to the increase in the lipophilic character of the metal chelate that favors its permeation through bacterial membranes. 

A series of metal complexes of Co(II), Ni(II) and Cu(II) have been obtained from the TCH derived from 8-formyl-7-hydroxy-4-methyl-coumarin (17, Scheme 6) [61]. This ligand has donor sites with the ONNO sequence and varied coordination possibilities: the metal complexes where characterized and evaluated for antibacterial (*Escherichia coli*, *Streptococcus aureus*, *Streptococcus pyogenes* and *Pseudomonas aeruginosa*) and antifungal activities (*Aspergillus niger*, *Aspergillus flavus* and *cladosporium*). The microbial results show that the Cu(II) complex shows high activity towards all the tested bacteria; Co(II) and Ni(II) complexes are weakly active. For antifungal activity, the Co(II) complex shows high activity against the two *Aspergillus* and less towards *Cladosporium*; the Ni(II) and Cu(II) complexes show high activity (almost equal to standard) in lower concentration towards *A. flavus*, and only moderate activity towards *cladosporium*.

Different coumarin derivatives show substantial cytotoxic activity in vitro and in vivo. Recently, the TCH derived from 3-acetylcoumarin was synthesized to construct a tetradentate ONNO donor ligand for the complexation of Co(II), Ni(II) and Cu(II) ions (18, Scheme 6) [62]. The compounds showed promising cytotoxic activity when screened using the in vitro method towards Ehrlich Ascites Carcinoma cells and, at the same time, were shown to have good activity when tested using the in vivo cancer model. It is worth mentioning that these effects were almost comparable to cisplatin used as the standard drug; however, the compounds were found to have good effect in prolonging the life span (ILS) as compared to cisplatin. The ligand and its complexes were also screened for antibacterial activity against *Escherichia coli*, *Staphylococcus aureus*, *Bacillus aureus* and *Aspergillus niger* and antifungal activity towards *Candida albicans*: the ligand is active against bacteria tested in the present study, but the promising results were observed for the copper complexes against all bacterial and fungal strains.

## 4. Macrocyclic Thiocarbohydrazones

The condensation reaction of a dihaldehyde/diketone with thiocarbohydrazide in a 1:2 stoichiometric ratio led to the formation of asymmetric TCH as structures 21 and 26 (see following section). When the stoichiometric ratio between the reagent is 1:1, it is possible to obtain the formation of polymeric structures [45,63] or to macrocyclic system capable to act as bidentate ligand towards metal ion [64,65]. In the latter case, acylferrocene undergoes an easy derivatization to form a macrocyclic bidentate system (19, Scheme 7) able to complex Co(II), Cu(II), Ni(II) and Zn(II) ions [66]. The antibacterial activity of these compounds was screened against *Escherichia coli*, *Bacillus subtillis*, *Staphylococcus aureus*, *Pseudomonas aeruginosa* and *Salmonella typhi*, and for antifungal activity against *Trichophyton longifusus*, *Candida albicans*, *Aspergillus flavus*, *Microsporum canis*, *Fusarium solani* and *Candida glaberata*. The ligand and the metal complexes possess good biological activity; however, a marked enhancement of activity was observed upon coordination with the metal ions against all the bacterial/fungal strains tested.

Macrocyclic TCH derived from 4,6-diacetylresorcinol (20, Scheme 7) and their metal complexes have been synthesized in two different ways: (i) a two-step procedure starting with the formation of the asymmetric TCH, followed by metal complexation and by reaction of the acyclic mono-nuclear VO(IV) and Ru(III) complexes with 4,6-diacetylresorcinol to afford the corresponding macrocyclic mono-nuclear VO(IV) and Ru(III) complexes; and (ii) a template reaction of the 4,6-diacetylresorcinol and thiocarbohydrazide with either VO(IV) or Ru(III) salts to afford the macrocyclic binuclear VO(IV) and Ru(III) complexes [67]. The ligands and the metal complexes were screened for their antimicrobial activity against *Staphylococcus aureus* as Gram-positive bacteria, and *Pseudomonas fluorescens* as Gram-negative bacteria in addition to *Fusarium oxysporum fungus*. According to the above reports, the ligand showed moderate biological activity against the tested strains and the complexes show higher bacterial activity as compared to the fungus, moreover the complexes of ruthenium showed the maximum inhibition against the growth of the selected bacteria and fungi.

## 5. Monothiocarbohydrazones and Asymmetric 1,5-Bisthiocarbohydrazones

Both hydrazine groups of thiocarbohydrazide display normal reactivity toward carbonyl compounds. Generally, the 1,5-diaddition products (Scheme 1a) are formed rapidly, and the monoadducts (Scheme 1b) are obtained under especially controlled conditions. In certain cases, however, there is a distinct difference in the reactivity of the first and second hydrazine groups of thiocarbohydrazide toward carbonyl compounds. The monothiocarbohydrazones (Scheme 1b) can be prepared through an effective method developed by Sandstróm: to a warm solution of thiocarbohydrazide in ethanol, in presence of acetic acid, an excess of the aldehyde or ketone is added; then, the product separates quickly on cooling [68]. In a simpler procedure, developed for the synthesis of a series of 2-acetylpyridine thiocarbonohydrazones, monoadducts have been prepared in large quantity, without contamination from possible bis-adducts, by treating the corresponding aldehydes and ketones with an equimolar amount of thiocarbohydrazide in methanol at reflux [28]. The synthetic procedures may involve a second step: the mono derivative can be reacted with a second aldehyde/ketone to achieve asymmetric 1,5-bisthiocarbohydrazones (Scheme 1c).

### 5.1. Asymmetric Thiocarbohydrazones Derived from Pyridyl and Related Compounds

One of the first reports on the biological activity of asymmetric pyridyl TCHs is dated 1956: this study by Brockman et al. reported that Pyridine-2-carboxaldehyde thiosemicarbazone and the corresponding thiocarbohydrazone consistently and significantly increased the life span of mice with L1210 leukemia [69]. As the for symmetric pyridil-based TCHs, only in the last decade have complexes with transition metals gained attention for their biological activity.

The TCH derived from 2,6-diacetylpyridine and its Mn(II), Co(II), and Cu(II) complexes (21, Scheme 8) were screened against the fungi *Alternaria brassicae*, *Aspergillus niger* and *Fusarium oxysporum* and the bacteria *Xanthomonas compestris* and *Pseudomonas aeruginosa* [70]. The complexes have moderate antipathogenic activities and are more effective than the free ligand; among the metal complexes, the Cu(II) derivatives are most active, probably due to redox processes and high affinity of these complexes to DNA of microorganisms.

In a comparison with symmetric TCHs (see previous section), quinolyl and pyridyl mono TCHs (22–24, Scheme 8) were tested for both their antitumor [52] and antimicrobial activity [54]. In the former report, mono-TCHs treatments induced considerably higher pro-oxidative activity compared to their bis-structural analogues; the increased pro-oxidant capacity was probably related to the terminal hydrazide group. Regarding the antimicrobial properties of the TCHs, the latter report is related to the kind of substituent at the nitrogen atoms: almost all of the mono TCHs show greater inhibition activity; the insertion of a methyl substituent at the 2-pyridyl derivative has the opposite effect: there is a decrease in the activity for mono TCHs, while it improves the activity of the symmetric derivatives against both Gram-positive and Gram-negative bacteria or fungi.

Water solubility is a great issue when dealing with TCHs ligand and complexes; in a recent report, the conjugation of an asymmetric pyridyl thiocarbohydrazone, as a metal delivery system with cytotoxic activity, and a glucose unit made it possible to ensure good water solubility at physiological pH (25, Scheme 8) [71]. The system enabled both complex zinc and copper metal ions to act as metal ionophores; the glycol-conjugation does not modify the cytotoxic activity of TCH unit on the human normal keratinocyte NCTC-2544, MDA-MB-231 breast cancer, or PC-3 human prostate adenocarcinoma cell lines; moreover, the biological activity is significantly higher than that reported for the 5-fluorouracil or cisplatin.

### 5.2. Asymmetric Thiocarbohydrazones Derived from Aromatics/Heteroaromatics, Isatin, Coumarin and Related Carbonyl Compounds

Medicinal chemistry has rarely taken advantage of ferrocene-containing compounds, nevertheless the ferrocene moiety has been combined with TCHs systems to achieve antibacterial/antifungal compounds and to further enhance the antibacterial/antifungal properties in coordination with transition metal ions such as cobalt(II), copper(II), nickel(II) and zinc(II) (26, Scheme 9) [72]. The ferrocenyl Tch and the metal complexes were all screened for their antibacterial activity against *Escherichia coli*, *Bacillus subtillis*, *Staphylococcus aureus*, *Pseudomonas aeruginosa* and *Salmonella typhi*, and, for antifungal activity against *Trichophyton longifusus*, *Candida albicans*, *Candida glaberata*, *Aspergillus flavus*, *Microsporum canis* and *Fusarium solani.* The compounds generally showed good antibacterial activity, but more significant antifungal activity was observed against most of the strains: the activity of the TCH increased by coordination of the metal ion. Generally, chelation/coordination reduces the polarity of the metal ion through delocalization of the positive charges within the whole chelate; these increases in the lipophilic character allow the metal complexes to go across the bacterial wall of microorganism and enhance the antimicrobial activity.

Mosa et al. synthesized and characterized the Cu(II), Ni(II), Co(III), Cr(III) and Fe(III) complexes of 4-hydroxy-coumarin derived TCH (27, Scheme 9); when screened for their microbiological activity, only the ligand and the cobalt complex showed significant activity for *C. albicans*, *E. coli* and weak activity against *P. aeruginosa* and *P. aeruginosa* [73].

In recent decades, a lot of asymmetric TCHs have been synthesized and successfully tested for their biological activity: as cytotoxic and/or chemopreventive agents with potent antiproliferative activity against cancer cells [74,75,76], for the antimicrobial activity against *B. subtilis*, *S. aureus*, *B. cereus* and *E. coli*, and antifungal efficacy over *C. albicans* and *A. fumigatus.* [76,77], for in vitro antiviral activity against various strains of DNA and RNA viruses [78], for evaluation of hepatoprotective activity [79], for the antidiabetic activity [80,81]. None of these TCH ligand systems have been evaluated in the presence of metal ion or as isolated metal complexes, making it clear that there is a lot of unexploited potential for this class of compounds as metal-based drugs.

## 6. Conclusions

Metal-based drugs are used in a wide spectrum of diseases as well as in medical diagnostics and represent a relevant sector of the pharmaceutical market. Accordingly, a multitude of novel ligands able to bind metal ions have been designed, synthesized and tested for their biological activity.

The thiocarbohydrazones (TCHs) are the homologue of thiosemicarbazones (TSCs), a class of compounds whose transition metal complexes have been extensively studied for their antimicrobial, antifungal and anti-cancer activities. The attractiveness of TCHs for developing new metal-based drugs lies primarily in the presence of extra metal binding motif so they can form both mononuclear and dinuclear complexes upon metal binding to a greater extent than TSCs. The asymmetry of TCHs can induce different metal coordination geometry so to tune more effectively the activity of redox active metals as copper that requires a change from a tetragonal to a tetrahedral ligand field. Furthermore, the extra hydrazine moiety of mono-TCHs is a reactive group for a further functionalization of the ligand to improve metal binding ability, solubility or cell membrane interaction. Surprisingly, despite the potential of TCHs as multifunctional ligands, not many studies have been carried out to evaluate biological activity of their metal complexes.

There are reports relating to the antibacterial and antifungal activity of thiocarbohydrazones complexes with different metal ions such as copper, nickel, zinc and a very limited number on the anticancer properties. Furthermore, few data concerning the molecular basis of the mechanisms underlying their activity are available.

The activity of TCHs metal complexes has been associated with chelating properties that are enhanced compared to that reported on the analogous complexes formed by thiosemicarbazones. On the other hand, the anticancer activity of both TCHs and TSC metal complexes may be associated with the ability of transport metal ions across cellular membrane acting as ionophore ligand so to induce an increase of intracellular oxidative stress. Different pathologies have been associated with metal dyshomeostasis, which can be the cause and/or effect of diseases ranging from cancers to neurodegenerative diseases [82,83]. A common line in these diseases is the presence of a strong excess of metal ions in the extracellular space and then the use of ligand able to chelate metal ions and restore a correct homeostasis represents an innovative therapeutic approach. With this aim, copper complexes with TSC have recently been used as potential drugs in neurodegenerative diseases such as Alzheimer’s Disease (AD) [84]. The metal complexes of TCHs have not yet been tested in AD models, but taking into account the enhanced properties to bind metal ions, they can represent a promising class of multifunctional drugs able to act not only as antibacterial, antifungal, anticancer but also to modulate metal homeostasis and then in perspective to be used in neurodegenerative diseases.

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
