# Peer review of "Biological Applications of Thiocarbohydrazones and Their Metal Complexes: A Perspective Review"

_pharmaceuticals, 2019, doi:10.3390/ph13010004_

Round 1
Reviewer 1 Report
The manuscript by Bonaccorso et al. reviews a topic on thiocarbohydrazones (TCH) and their metal complexes, together with their (potential) biological applications. As little is indeed known at the moment on these TCHs in comparison with their thiosemicarbazones (TSC) homologues, this short review is definitely timely and of interest to the readership.
In general, the manuscript is written in a proper and structured way.
I would suggest that the authors try to comment more on the mechanism of action (if possible of course) of several of the complexes mentioned. Now this seems quite limited and this would in my opinion be of added value to the work. E.g. r183, is there more information known on the DNA cleavage mechamism?
In that regard, it would be a good idea to summarise/comment shortly on the reported mechanism(s) of action of such TCHs in a separate paragraph?
r232: what's the rational behind the improved interaction of the trans-isomers?
Why mentioning about AD in the conclusions? Please then dedicate a paragraph on the use of TCHs in neurodegenerative diseases as well?
Furthermore, I have some more minor comments on the manuscript:
r39: please make sure you write the names of species like Helicobacter pylori correctly
r57: please make sure you that the use of capital letters is correctly used (e.g. 'thiosemicarbazones' instead of 'Thiosemicarbazones'). Further on, e.g. drugs like vancomycin, streptomycin are written with capitals as well? Please check the whole manuscript for similar issues (r65, r177, r178, r421).
Why writing aldehyde/ketone with capitals? Please check the manuscript.
r75: sentence is not clear?
r122, r427: TCH instead of Tch?
r125: members instead of member
r129: can you refer to the proper references please?
r194: 'the tested' instead of 'tested'
r207, r208: 'pyridyl' instead of 'pyridil'
r265: pyridyl instead of Pyridyl
r270: ketones instead of Ketones
r301: 'N-methylisatin' instead of 'N-Methylisatin'
r338, r339, r417: cisplatin instead of Cisplatin
r353: acylferrocene instead of Acylferrocene
r390: reports instead of report
r409: 'G positive'? 'G negative'?
r417: fluorouracil instead of fluorouracile
r423: taken instead of take
r445: antidiabetic?
r480: have instead of has
Author Response
Reviewer 1
The manuscript by Bonaccorso et al. reviews a topic on thiocarbohydrazones (TCH) and their metal complexes, together with their (potential) biological applications. As little is indeed known at the moment on these TCHs in comparison with their thiosemicarbazones (TSC) homologues, this short review is definitely timely and of interest to the readership.
In general, the manuscript is written in a proper and structured way.
We thank the referee for the positive comment.
I would suggest that the authors try to comment more on the mechanism of action (if possible of course) of several of the complexes mentioned. Now this seems quite limited and this would in my opinion be of added value to the work. E.g. r183, is there more information known on the DNA cleavage mechamism?
Although the authors report clear evidences on the DNA cleavage, there are no specific details on the cleavage mechanism. We have modified the text reporting the suggested general mechanism: “This cleavage activity is related to the ability of the complexes to diffuse into bacterial cytoplasm, thanks to the increased membrane fluidity (the chelation of the metallic cations exalts the hydrophobicity of the systems, the subsequent insertion of complexes into bilayer membrane affect lipid packaging). The non-covalent interactions of the metal complexes with DNA (intercalation and groove binding) cleave the duplex, cause DNA damage, block the division of cells and result in cell death.”
In that regard, it would be a good idea to summarise/comment shortly on the reported mechanism(s) of action of such TCHs in a separate paragraph?
There are no many data reported on the mechanism of action of TCHs. However, following reviewer’s suggestion, we have inserted a short paragraph at the end of the Introduction section and expanded the text into the following sections.
“For the same reasons, together with the novelty represented by these systems and the numerous fields of application, it is not possible to identify a unique action mechanism: each combination (metal cation / TCHs backbone / field of employ) can give rise to a different mechanism of action which, only in few cases, has been deeply investigated; when available, clear indications of the action mechanism of the TCHs metal complexes have been reported in the text.”
r232: what's the rational behind the improved interaction of the trans-isomers?
According to reviewer’s suggestion, we added the following paragraph:
“The Ni(II) and Co(II) complex showed the best interaction with DNA due to both the possible geometry of metal coordination to the TCH ligand (distorted tetrahedral in the former case and square pyramidal in the latter, they were indicated as trans-geometrical isomers) and the availability of the vacant d-orbital. The strong stacking interaction between the aromatic chromophore and the base pairs define the DNA binding property of the metal complexes: the Ni(II) and Co(II) complexes acts as groove binder with the DNA. The Cu(II) and Zn(II) complexes have much less binding affinity; they even have different binding modes of interaction or bonding to the DNA.”
Why mentioning about AD in the conclusions? Please then dedicate a paragraph on the use of TCHs in neurodegenerative diseases as well?
At the best of our knowledge there are no studies on the use of TCHs in neurodegenerative diseases. Instead there are data on thiosemicarbazone derivatives activity in Alzheimer’ s models. Taking into account the strong analogy between TCHs and TSC, in the conclusion section we mentioned the AD only as future potential target of TCHs. We clarify better this point in the revised version.
Furthermore, I have some more minor comments on the manuscript:
r39: please make sure you write the names of species like Helicobacter pylori correctly
This typo has been corrected
r57: please make sure you that the use of capital letters is correctly used (e.g. 'thiosemicarbazones' instead of 'Thiosemicarbazones'). Further on, e.g. drugs like vancomycin, streptomycin are written with capitals as well? Please check the whole manuscript for similar issues (r65, r177, r178, r421).
The text has been checked and made uniform
Why writing aldehyde/ketone with capitals? Please check the manuscript.
The manuscript has been corrected
r75: sentence is not clear?
The sentence has been rewritten
r122, r427: TCH instead of Tch?
These typos have been corrected
r125: members instead of member
This typo has been corrected
r129: can you refer to the proper references please?
The reference has been revised
r194: 'the tested' instead of 'tested'
This typo has been corrected
r207, r208: 'pyridyl' instead of 'pyridil'
These typos have been corrected
r265: pyridyl instead of Pyridyl
This typo has been corrected
r270: ketones instead of Ketones
This typo has been corrected
r301: 'N-methylisatin' instead of 'N-Methylisatin'
This typo has been corrected
r338, r339, r417: cisplatin instead of Cisplatin
These typos have been corrected
r353: acylferrocene instead of Acylferrocene
This typo has been corrected
r390: reports instead of report
This typo has been corrected
r409: 'G positive'? 'G negative'?
These typos have been corrected
r417: fluorouracil instead of fluorouracile
This typo has been corrected
r423: taken instead of take
This typo has been corrected
r445: antidiabetic?
This typo has been corrected
r480: have instead of has
This typo has been corrected

Reviewer 2 Report
The article entitled: "Biological Applications of Thiocarbohydrazones and Their Metal Complexes: a perspective review"
Is meant to describe synthesis of thiocarbohydrazone ligands and their complexes thereof followed by systematic biological applications.
The paper was written very casually with serious language issues. Moreover, there are also some scientific errors.
The article do not mech biological applications which are mentioned sporadically.
Moreover, it is very difficult to follow the structures of the corresponding complexes which are sporadically mentioned and not organized systematically.
The manuscript is not mature for the publication, if at all, in Pharmaceuticals. Main topic, so far, is synthesis of thiocarbohydrazone ligands and complexes thereof for which the structure is very difficult to deduce.
Author Response
Reviewer 2
The article entitled: "Biological Applications of Thiocarbohydrazones and Their Metal Complexes: a perspective review"
Is meant to describe synthesis of thiocarbohydrazone ligands and their complexes thereof followed by systematic biological applications.
The paper was written very casually with serious language issues. Moreover, there are also some scientific errors.
Following reviewer’s suggestion, these typos have been corrected. Also, the manuscript has been improved in the organization following the suggestion of the referees.
The article do not mech biological applications which are mentioned sporadically.
Following reviewer’s suggestion, we have inserted a short paragraph at the end of the Introduction section and expanded the text into the following sections.
“For the same reasons, together with the novelty represented by these systems and the numerous fields of application, it is not possible to identify a unique action mechanism: each combination (metal cation / TCHs backbone / field of employ) can give rise to a different mechanism of action which, only in few cases, has been deeply investigated; when available, clear indications of the action mechanism of the TCHs metal complexes have been reported in the text.”
Furthermore, highlighting the considerations already included in the first version of the ms, here we try to resume and briefly summarize the existing literature concerning TCHs, poorly investigated so far in their medicinal applications. Considering as the progress in the design, synthesis and preparation of novel metal-based drugs, largely overlaps the selection of the best ligands; we think that this kind of report, focusing on this class of overlooked ligands/complexes, may be of high interest for people working in different fields of drugs development, ranging from chemistry to biology, pharmacology and biochemistry.
Moreover, it is very difficult to follow the structures of the corresponding complexes which are sporadically mentioned and not organized systematically.
It is extremely hard -if not impossible-to categorize and organize all the possible structures of TCHs complexes in view of the extreme versatility of this family of ligand. Indeed, one ligand may originate metal-complexes with several geometries strictly dependent on the synthesis conditions, the different heteroatoms that are present in the structures and several other aspects. Thus, in our opinion to ensure readability of the ms the best organization is that based on the backbone of the ligand. We explain this point in the text as follow:
“Due to the huge variety of metal cations employed in biologically active TCHs complex (eighteen cationic species, ranging from the first-row transition metal cation to lanthanides, from organo-metal to oxo-metal cation, will be reported in this review) and to the different binding stoichiometries available for each cation, we decided to employ a simpler classification based on the ligand backbone: the biological activity of the metal complexes derived from symmetric, macrocyclic and asymmetric TCHs has been discussed in three different section. Further sub-sections have been employed to describe the most common substituents used in the TCHs ligand synthesis (R1, R2, R3 and R4 groups in Scheme 1), namely salycilic, pyridyl, aromatic/heteroaromatic, ferrocenyl, isatin, coumarin, (and related) carbonyl compounds.”
Moreover, in each sub-section the different systems have been reported following a mere chronological order.
The manuscript is not mature for the publication, if at all, in Pharmaceuticals. Main topic, so far, is synthesis of thiocarbohydrazone ligands and complexes thereof for which the structure is very difficult to deduce.
We respect the position of this referee, but we disagree. Again, we believe that the synthesis and design of innovative and more effective drugs encompasses the choose and evaluation of the more convenient/effective ligands for the intended medicinal purposes. This means that the knowledge and the understand of the various families of ligands is extremely important. Indeed, beyond the functional metal center that exerts the pharmacological activity, this activity can be conveniently tuned and modulated by choosing the best ligand for any specific medicinal application. Thus, it is clear, that at variance with TSCs, TCHs are far less common and have been scarcely investigated. Yet, TCHs are very versatile ligands that can bind the metal centers through different coordination sites (they can act as multidentate ligands) and different geometries. The precise description and understanding of their chemistry, alongside the knowledge about their potential in medicinal applications, remain largely unmet. In this view our review posses a two-fold significance: on one hand, reports on the feasibility concerning the synthesis of TCHs-based metal complexes for medicinal purposes; on the other hand, it summarises the main features and characteristics of this family of ligand providing to researcher a valid tool in order to evaluate TCHs as ligands for metal complexes suitable in several field of medicinal chemistry.

Reviewer 3 Report
This manuscript is focused on the literature of thiocarbohydrazones and their metal complexes as biological agents. The authors have conducted a thorough literature review, undertaken a rigorous piece of data collection and have analyze information accurately. I believe that this paper is suitable to publication in Pharmaceuticals, however the authors should address the following minor points
1. Lines 57-63. This paragraph is quite confusing, specially in the following points a) “they can act as multi-dentate ligands due to the available nitrogen and it is easily to modify the backbone” this sentence needs to be rewritten in order to include the sulfur atom as well as others heteroatoms from the backbone structure and b) reference 12 refers only to anticancer activity of thiosemicarbazones however thiosemicarbazones have been extensively studied as antiviral, antibacterial and antifungal agents and therefore more references should be included.
2. Line 90. In scheme 1, structure (d) is not mentioned in the text, should be removed
3. Line 459. “TSCs form mainly mononuclear complexes”. This is not correct since one of the main characteristics of these compounds is their ability to form di- and poly-nuclear metal complexes.
Author Response
Reviewer 3
This manuscript is focused on the literature of thiocarbohydrazones and their metal complexes as biological agents. The authors have conducted a thorough literature review, undertaken a rigorous piece of data collection and have analyze information accurately. I believe that this paper is suitable to publication in Pharmaceuticals, however the authors should address the following minor points
We thank the referee for the positive comments.
Lines 57-63. This paragraph is quite confusing, specially in the following points a) “they can act as multi-dentate ligands due to the available nitrogen and it is easily to modify the backbone” this sentence needs to be rewritten in order to include the sulfur atom as well as others heteroatoms from the backbone structure
The text has been modified following reviewer suggestions.
and b) reference 12 refers only to anticancer activity of thiosemicarbazones however thiosemicarbazones have been extensively studied as antiviral, antibacterial and antifungal agents and therefore more references should be included.
The reference has been updated following reviewer suggestion.
Line 90. In scheme 1, structure (d) is not mentioned in the text, should be removed
The structure (d) is now mentioned in the text (see Introduction section, line 64-66)
“Beyond TSCs, in the last years, there has been growing interest in the coordination chemistry of thiocarbohydrazones (TCHs) compounds that share the general formula depicted in Scheme 1a-c and that can be considered the higher homologues of TSCs (Scheme 1d).”
Line 459. “TSCs form mainly mononuclear complexes”. This is not correct since one of the main characteristics of these compounds is their ability to form di- and poly-nuclear metal complexes.
The sentence was misleading and has been rephrased accordingly.

Round 2
Reviewer 2 Report
There has been not much of improvement of the manuscript in the direction of Biological application.
The article deals mostly the synthesis of ligands.
No appropriate schemes showing the structures of the corresponding complexes of Thiocarbohydrazones was added and there biological activities are not comprehensively discussed as mentioned in the title of the manuscript. “Biological Applications of Thiocarbohydrazones and Their Metal Complexes: a perspective review”